# I Do Not Want to Miss a Thing! Consequences of Employees’ Workplace Fear of Missing Out for ICT Use, Well-Being, and Recovery Experiences

**DOI:** 10.3390/bs14010008

**Published:** 2023-12-22

**Authors:** Linda-Elisabeth Reimann, Carmen Binnewies, Phillip Ozimek, Sophie Loose

**Affiliations:** 1Department of Work Psychology, University of Muenster, 48149 Muenster, Germany; carmen.binnewies@uni-muenster.de (C.B.); sophie.loose@uni-muenster.de (S.L.); 2Department of Clinical Psychology and Psychotherapy, Ruhr University Bochum, 44801 Bochum, Germany; phillip.ozimek@ruhr-uni-bochum.de

**Keywords:** workplace fear of missing out, information and communication technologies, recovery experiences, work-related well-being, perceived stress, burnout

## Abstract

As more and more employees have access to work-related information and communication technologies (ICTs) anywhere and anytime, new challenges arise in terms of well-being and recovery experiences. Feelings of workplace fear of missing Out (wFoMO) and workplace telepressure may be personal demands that add to the literature of the job demands-resources (JD-R) theory. In this study, we proposed a model in which wFoMO and workplace telepressure were associated with employee well-being variables via the use of ICTs during leisure time. Therefore, we analyzed the data of *N* = 130 employees who answered two questionnaires in the interval of one work week. The results revealed negative indirect effects between wFoMO/workplace telepressure and psychological detachment/perceived stress via ICT use. The results were more ambivalent regarding the dependent variables burnout, relaxation, and control. This strengthens the literature that categorized ICT use as a job demand. However, we also found positive indirect effects on perceived informational benefits, which supports the idea of ICTs being both a job demand and a job resource in light of the JD-R theory. This study contributes to past research on work-related ICT use during leisure time and demonstrates the relevance of personal demands such as wFoMO for employees’ well-being.

## 1. Introduction

The way we work and live has drastically changed since digitalization- and technology-mediated communication became ubiquitous in nearly every life domain [1]. Interactions at work are no longer confined to the workplace but have found their way into the place most people use for recovery—their home [2,3]. We already know that this shift has negative consequences regarding work–life balance, boundary management, and overall employee well-being [4,5,6,7]. Past research has also connected the construct of workplace telepressure—the urge to quickly respond to messages—with the need to use information and communication technologies (ICTs) for work interactions [8]. Other studies have perpetuated this research and found that the experience of workplace telepressure is a crucial factor in predicting ICT use during leisure time and has negative consequences for employee psychological detachment [9]. In this study, we aimed to extend past evidence by considering another individual variable named workplace fear of missing out (wFoMO) [10]—the fear of missing out work-related information. We argued that wFoMo is essential for explaining why ICTs are used during non-working times and influence employee well-being. The relatively new construct of wFoMO has its origins in media psychology, where the term fear of missing out (FoMO)—the fear of missing out on rewarding experiences—is well known when it comes to private media engagement [11]. We derived our hypotheses from empirical evidence regarding workplace telepressure, general FoMO, and work-related ICT use, as well as by considering one of the most prominent stress frameworks, the job demands-resources theory (JD-R theory) [12]. We examined whether the variables wFoMO and workplace telepressure, mediated by ICT use, influence key work-related outcomes, namely positive and negative effects, perceived burnout, and recovery experiences (psychological detachment, relaxation, and control) after a work week. We additionally assessed the outcome variable of perceived informational benefits to further explore the possible positive effects of ICT use after work. To the best of our knowledge, this study is the first to assess wFoMO and workplace telepressure together in a study with two paired measurement points and to investigate more detailed the possible consequences that these constructs may have on employee well-being via ICT use. Our study provides three major contributions: (1) integrating wFoMO and workplace telepressure as personal demands in JD-R theory, (2) being one of the first studies to explore how these personal variables affect employee well-being in combination with after-workhour ICT use, and (3) helping to shed light on the question of whether ICT use can solely be perceived as a job demand vs. a job resource (i.e., investigating negative vs. positive effects from ICT use on employee well-being). This raises the overall research question of whether wFoMO and workplace telepressure are important predictors for employee well-being via ICT use. To address this question, we will first explain the origin of wFoMO and its relationship with workplace telepressure and ICT use. We will then derive our hypotheses by referring to the job demands-resources theory in combination with the most important empirical evidence. This evidence is currently limited, particularly regarding the relatively new construct of wFoMO.

### 1.1. Theoretical Background

In media research, the term fear of missing out (FoMO) became quite popular. FoMO is the pervasive apprehension that one might miss rewarding experiences compared to their friends or relatives [11]. Since the prevailing work of Przyblinski and colleagues, many studies have been conducted for a better understanding of the origins and consequences of FoMO [13]. Most of these investigations focused on the consequences of FoMO on different well-being variables by also considering individuals’ social media engagement [13,14]. In this context, a meta-analysis focusing on private FoMO found that social media engagement can be both an antecedent and consequence of experiencing FoMO [15,16,17,18,19]. However, media usage is not limited to people’s private spheres but also becomes crucial in the context of many working environments [20]. For example, 60% of employees stated that they used mobile devices for their work in the year 2022 [21]. Due to these developments, it would be insufficient to not also consider the consequences of FoMO on the professional sphere of employees.

### 1.2. Workplace Fear of Missing out vs. Fear of Missing out at the Workplace

When investigating FoMO in professional contexts, it is important to distinguish between two kinds of approaches in past studies. To our knowledge, most studies on FoMO in such professional contexts focused on what we would call fear of missing out at the workplace, meaning that employees may be prevented from doing their work properly due to their private feelings of FoMO [22]. Investigations that used this focus explored whether outcome variables such as work performance decrement, procrastination, job performance, or productivity at work would be affected with private social media engagement and feelings of FoMO during working hours [22,23,24]. However, FoMO in professional contexts can also be conceptualized as employees experiencing the fear of missing out on something that relates to work itself. We only found one other study conducted by Budnick et al. [10] that described this phenomenon of workplace fear of missing out (wFoMO). In their study, wFoMO was conceptualized as the constant fear that employees may miss beneficial work-related information or career opportunities when physically absent from work [10]. In line with this, they also validated a questionnaire to assess wFoMO that distinguishes between the dimensions of perceived informational and relational exclusion. While the first aspect relates to the fear of missing out on valuable work information, the latter describes the fear of losing out on networking opportunities or not maintaining important business contacts that other colleagues may have [10].

### 1.3. Workplace Fear of Missing out and ICT Use

When researching FoMO, it is necessary to acknowledge that, in addition to ICT use, social media engagement also plays an important role in this context [13]. However, how FoMO and social media engagement are expected to associate varies within studies, depending on the tackled questions and their underlying theoretical foundations [13]. Several studies have defined FoMO as a trait and showed that it leads to more social media engagement and, in turn, to further outcomes such as maladaptive social media behaviors [17] or harmed individual well-being, such as stress or life satisfaction [8,25]. Other studies have showed that social media engagement itself reinforces FoMO, and this negative state of mind leads to further problematic outcomes [26,27]. In fact, the majority of available research seems to agree that FoMO is strongly connected with social media engagement. However, it does not necessarily mean that this association also holds for wFoMO. Budnick et al. [10] revealed that wFoMO and social media engagement show a small positive correlation. Although professional social media sites like LinkedIn were used by 242 million people in Europe in 2023 [28], they may not be the most crucial variable for everyday experiences of regular employees. Instead of social media engagement, the variable of information and communication technologies (ICTs) has often been utilized when it comes to digital technology engagement in the workplace [29]. In this study, we used the definition by Ratheeswari, who defined ICTs as all kinds of technologies that “provide access to information though telecommunication” [30]. A special feature of ICTs is that it allows for employees to stay connected to work even when their regular working hours are already over [20]. We argued that employees who experience higher levels of wFoMO are more inclined to use ICTs during leisure time to reduce the feeling of missing out on something. Similar findings have already been documented in studies on general FoMO [14]. They argued that the experience of FoMO is due to the lack of satisfaction of the need for relatedness/need to belong, which individuals hope to reduce by using social media. From a theoretical perspective, the same mechanism could also hold for wFoMO, i.e., that people with higher levels regarding the need for relatedness also experience more wFoMO and are constantly afraid that their colleagues do not include them. Otherwise, rather, performance-related needs may be the origin of experiencing wFoMO, which would provide a strong distinction from private FoMO experiences. However, there is a lack of evidence regarding whether such effects also hold for wFoMO [13]. Therefore, we included workplace telepressure in this study, which has been more established in the field of work psychology.

### 1.4. Workplace Telepressure

In addition to wFoMO, workplace telepressure has been defined as a state of mind in which employees feel a strong preoccupation and urge to respond quickly to work-related messages [8]. It has been revealed that both constructs, wFoMO and workplace telepressure, are positively associated with one another but still represent distinct constructs [10,31]. The same has also been found for private FoMO and telepressure in student samples [8,32], for which it has also become clearer as to why both constructs must be distinguished. For example, Rogers and Barber [32] found that although both constructs were positively associated with social media engagement, only telepressure was linked to technology use during sleep (i.e., checking messages on the smartphone in micro-breaks during sleep). Workplace telepressure has been further described “as a psychological state that encourages people to stay connected to work-related communications via information and communication devices” [8]. In this definition, it is evident that the perception of workplace telepressure requires the existence of ICTs. It is still an open question as to whether this also holds for wFoMO, which will be tackled in this investigation.

Furthermore, empirical evidence has suggested that workplace telepressure can harm employee well-being via the usage of ICTs [9,33]. However, while several studies argued that direct effects can be expected between workplace telepressure and well-being outcomes [31,34], others stated that ICT use is necessary for negative effects to emerge [35,36]. Although this evidence may appear contradictory, it is important to note that studies used different conceptualizations and operationalizations (e.g., some focused on ICT use during leisure time, while others focused on general ICT use). Nevertheless, all studies seem to agree that ICT use needs to be considered as a crucial variable when investigating workplace telepressure effects. In our study, we specifically focused on ICT use during leisure time. This directly led to the research question of this study of whether wFoMO and workplace telepressure may have negative consequences on employee well-being when using ICTs during non-working times. To address this question and explain why we defined ICT use as a central mediator variable, we drew on the job demands-resources theory (JD-R theory) [12] as a central stress model in work psychology.

### 1.5. The Job Demands-Resources Theory

One of the most influential frameworks in work psychology is the JD-R theory. The JD-R theory explains how personal and job-related aspects can not only lead to positive and negative outcomes for work engagement but also work-related well-being [37,38]. This theory has been applied to many occupational settings [39] and states that different job characteristics can be classified as either a job demand or a job resource [40]. The use of ICTs during leisure time is often seen as a typical job demand, as it is associated with costs like feelings of technostress, information overload, and blurred work–life boundaries [40,41,42,43]. Past research has also shown that the usage of ICTs outside regular working hours can hinder employees from adequately recovering from their work [5,44].

However, it is possible that the use of ICTs does not necessarily lead to these negative effects for all employees. Extensions of the JD-R theory describe that, alongside job demands and resources, personal demands and resources must also be considered [38]. WFoMO and workplace telepressure may be seen as such personal demands. Individuals may experience distress when feelings of wFoMO or workplace telepressure occur. For general FoMO and workplace telepressure, this association has already been found in past research [8,25]. In this context, subsequent ICT use is considered a coping mechanism where employees hope to reduce individual distress caused by wFoMO and telepressure. Therefore, and based on the empirical evidence [8,10], we assumed the following:

**Hypothesis 1a-b:** 
*WFoMO (H1a) and workplace telepressure (H1b) are positively associated with ICT use during leisure time.*


In the following sections, we present key work-related well-being variables that are assumed to be negatively affected by the interaction of personal and job demands (e.g., wFoMO, workplace telepressure, and ICT use). However, we will also explain why this process should enhance perceived individual benefits [45], leading to a more comprehensive view on the role of work-related ICT use as a job demand vs. a job resource.

### 1.6. Consequences of wFoMO, Telepressure, and ICT Use

Several recent studies investigated whether and how ICT use affects employee well-being. In this context, research ranges from studies that assessed ICT use during a typical workday [46] or during leisure time [47] to studies that examine the effects of ICT use on work–life boundaries or work–life balance [48]. Many of these investigations also relied on the JD-R theory to substantiate their assumptions [49,50]. In the current study, we investigated how three central work-related well-being variables are affected by the interaction of wFoMO, workplace telepressure, and ICT use, namely feelings of *perceived stress*, *burnout*, and *recovery experiences* (psychological detachment, relaxation, and control, respectively). These variables were selected as they have often been used as indicators of employee well-being [34,36,44,48,50].

According to the JD-R theory, such job-related well-being variables decline when job demands are high [39]. ICT use, therefore, becomes a stressor if employee resources are compromised, which is likely to be the case with leisure-time ICT use and the presence of additional personal demands, such as wFoMO and workplace telepressure. Past research has shown that ICT use during leisure time can enhance stress levels [44,51]. Furthermore, burnout is another indicator of work-related well-being that is of particular interest. Burnout is defined as a psychological syndrome that emerges due to stressors on the job [52]. Past studies have revealed that burnout is a central variable that can be negatively affected by high job demands [53,54,55]. As described above, ICT use during leisure time can enhance perceived stress due to additional personal demands, which can, in turn, reduce work well-being [42,50]. An investigation by Derks and Bakker [44] found that feelings of burnout are negatively affected by after-workhour ICT use. This finding is strengthened by a recent investigation that stated that ICTs are important in the development of burnout [48]. Therefore, we assumed that the following:

**Hypothesis 2a-b:** 
*Using ICTs during leisure time is positively associated with perceived stress (H2a) and burnout (H2b).*


When looking at the literature on the consequences that job demands can have on employees, recovery is ubiquitous. Recovery from work refers to processes that reduce perceived strain due to prior demands [56]. In this context, the term of recovery experiences [57] is often used “to characterize attributes associated with off-job activities contributing to recovery” [3]. In this study, we focused on the recovery experiences *psychological detachment*, *relaxation*, and *control*. On the one hand, psychological detachment and relaxation need to be considered, as both recovery experiences should be achieved during leisure time when typically no job demands are present and employees aim to adequately recover from their work [3]. However, with the new possibilities to work from nearly everywhere using ICTs [4] and the omnipresence of perceptions such as wFoMO and workplace telepressure, the achievement of psychological detachment and relaxation during leisure time may become less frequent [31]. On the other hand, we argue that control is also essential to consider as a recovery experience. When employees use ICTs after work due to feelings of workplace telepressure and/or wFoMO, feelings of control are harmed [2], especially if they do not want to engage in ICT use but feel forced to do so, receive push notifications, or perform it unconsciously as they receive messages on private devices.

**Hypothesis 2c:** 
*Therefore, we hypothesized that using ICTs during leisure time is negatively associated with recovery experiences, namely psychological detachment, relaxation, and control.*


As previously stated, ICT use may not solely be perceived as a job demand but also as a job resource. A study by Ninaus et al. [48] made a similar argument and showed that the possibility of using ICTs for work was perceived as rather positive. Despite that, the consequences regarding ICT demands were reported as more intense than the positive aspects resulting from ICT resources [48]. While Ninaus et al. [48] assessed ICT use during working hours, we specifically looked at ICT use during leisure time. Therefore, it may be that in trying to adequately demonstrate the positive effects of ICT use after work, other variables than those that have already been examined need to be considered. The perception of *informational benefits* may be such a variable. These informational benefits can be achieved when employees share work-related knowledge with each other, as well as when colleagues help each other by gaining more job security or career advancements [58]. Past research has revealed that the usage of professional social media sites, like LinkedIn, results in higher levels of informational benefits [45]. It could be that this finding also applies to general work-related ICT use, as continuous exchanges with colleagues or people working in the same field could give the impression of having greater work-related informational benefits. To our knowledge, this construct has not been assessed in a similar study design as an indicator of the positive effects of after-workhour ICT use. In the light of this evidence and the JD-R theory, we assumed the following:

**Hypothesis 2d:** 
*Using ICTs during leisure time is positively associated with informational benefits.*


Despite the extensive evidence regarding the consequences of ICT use as a job demand, or in some cases as a job resource, less is known about the roles of wFoMO and workplace telepressure as personal demands. However, recent research has shown that workplace telepressure may be a crucial factor when it comes to the question of whether employees perceive work-related ICT use as a demand [59]. Moreover, there is empirical evidence that workplace telepressure as a personal demand does not have a direct effect on variables such as psychological detachment [9] or overall psychological resource depletion [33,60] but an indirect effect via ICT use [9,33]. Furthermore, when considering the literature on general FoMO, the relationship between FoMO and social media engagement appears to be one of the most important factors when exploring the negative consequences for individuals [13]. In the context of media theories [61,62], FoMO is simply a state of mind arising from the essential need to belong/need for relatedness [11]. However, when using social media as a maladaptive and dysfunctional coping mechanism, negative consequences can occur [62]. In this study, the same was hypothesized for wFoMO.

Therefore, we did not propose direct effects between wFoMO as well as workplace telepressure and perceived stress, burnout, recovery experiences, and informational benefits, but rather suggest indirect effects with ICT use during leisure time as the crucial mediator variable. We argued that work-related well-being is not harmed by the experience of wFoMO and telepressure alone but rather by the possibility to engage in maladaptive coping mechanisms, specifically ICT use during non-working times. Moreover, ICTs are essential for experiencing workplace telepressure, and we aimed to shed light on the open question of whether this also applies to wFoMO. In addition, we only expected indirect effects between wFoMO and workplace telepressure with informational benefits via ICT use. The reasoning behind this is similar: wFoMO and telepressure are emotional states (or personal demands) that lead to ICT use. However, they cannot lead to the expected outcomes without engagement in ICT use. Therefore, we hypothesized the following:

**Hypothesis 3a-c:** 
*There are negative indirect effects between wFoMO and the variables perceived stress (H3a), burnout (H3b), and recovery experiences (H3c) mediated by using ICTs during leisure time.*


**Hypothesis 3d:** 
*There is a positive indirect effect between wFoMO and informational benefits mediated by using ICTs during leisure time.*


**Hypothesis 4a-c:** 
*There are negative indirect effects between workplace telepressure and the variables perceived stress (H4a), burnout (H4b), and recovery experiences (H4c) mediated by using ICTs during leisure time.*


**Hypothesis 4d:** 
*There is a positive indirect effect between workplace telepressure and informational benefits mediated by using ICTs during leisure time.*


See also Figure 1 for a better understanding of these hypotheses (note that these hypotheses also represent the overall research question). To summarize, with this study, we will contribute to the question of whether ICT use can be considered as a job demand vs. a job resource in the context of the JD-R theory [48]. By doing so, we extend prior research by establishing the relatively new construct of wFoMO as well as workplace telepressure as personal demands that may become essential to consider due to the influence of ICTs in work contexts and, therefore, usage patterns during leisure time [50].

## 2. Methods

The current study has been preregistered at the Open Science Framework (OSF; https://osf.io, accessed on 9 September 2023) and is available at https://doi.org/10.17605/OSF.IO/B2PVW (accessed on 9 September 2023). The data used in this study were part of a larger project (see https://doi.org/10.17605/OSF.IO/YXTVZ for an overview of the whole project, accessed on 9 September 2023). The ethics committee of the Psychology and Sports Department of the University of Muenster has previously checked and approved the safety of the procedure. Data handling and data management were organized in accordance with the DFG guidelines and their specification by the German Psychological Society. The materials, data, and scripts that were used in this study are also available at the OSF via https://doi.org/10.17605/OSF.IO/YXTVZ (accessed on 9 September 2023).

### 2.1. Procedure

The data collection procedure consisted of two measuring points with an interval of one work week. The participants were made aware of this study through the distribution of handouts and digital flyers. To address possible participants, flyers were distributed, e-mails were sent, and social media sites like LinkedIn, Instagram, and WhatsApp were utilized. Each participant started the first questionnaire on a Monday (t1) of a regular working week and was reminded after six days (t2) to answer the second questionnaire on the following weekend. The questionnaires were sent via e-mail, which was possible due to the prior registration of the participants for this study. At t1, all variables were assessed on a general level, whereas at t2, we specifically asked how the participants felt during their last work week. For data collection, formr (LINK) software was utilized.

### 2.2. Participants

The data were collected by recruiting employed participants of all occupational fields. To address possible participants, flyers were distributed, e-mails were sent, and social media sites like LinkedIn, Instagram, and WhatsApp were utilized. One hundred and fifty-seven participants completed the survey at t1, and only one hundred and thirty-one participants completed it at t2. Additional attrition analyses showed that 34.73 percent of the participants who had started the first questionnaire did not complete it. From t1 to t2, an additional 13.38 percent of the participants who had completed the first questionnaire dropped out. Furthermore, we compared all the variables and demographic data at t1 and t2 by conducting *t*-tests. The results showed that the sample has not statistically changed in terms of its demographic information or most of the assessed constructs. However, significant differences were revealed for informational wFoMO (*t*(268) = 2.08, *p* = 0.037), relational wFoMO (*t*(282) = 3.05, *p* = 0.002), and informational benefits (*t*(263) = 2.30, *p* = 0.022). These variables were rated significantly lower at t2 compared to t1, meaning that people with high wFoMO and informational benefits were underrepresented. Only the participants that completely answered both surveys were included in the analyses, resulting in a final sample of 130 participants with 50 males and 80 females. The mean age was 37.96 years (*SD* = 12.70), and the sample was highly educated, with 47.62 percent of the participants having an academic degree. Twenty percent of the participants completed their A-levels, and only 5.38 percent reported having a lower qualification (see also Appendix A). The mean duration of weekly working hours was 34.88 h (*SD* = 10.79), and 37.69 percent of the participants currently held a management position.

### 2.3. Measures

The same variables were assessed at t1 and t2. However, as stated above, the instructions regarding how to answer the items slightly varied. In addition to this, the collection of demographic variables was only carried out once at t1. The demographic variables included age, gender, job security, level of education, current job, living situation, number of children, whether a leading position was held, and how many hours each individual worked per week.

All scales were answered on a five-point Likert scale (ranging from 1 = “do not agree” to 5 = “totally agree”) if not stated otherwise. If a German version was not available for specific scales, we utilized the “forward-backward procedure” [63] for translation.

**Workplace fear of missing out**. For assessing wFoMO, a questionnaire developed by Budnick et al. [10] was utilized. This scale consists of ten items that are divided into two dimensions: *informational* (e.g., I worry that I might miss important work-related updates) and *relational exclusion* (e.g., I am constantly thinking that I might miss opportunities to strengthen business contacts) at the workplace. Budnick et al. [10] reported that both dimensions explain an 87 percent variance in wFoMO and show excellent reliability coefficients (relationship exclusion α = 0.92; information exclusion α = 0.93). In this study, reliability coefficients for the wFoMO scale were also excellent, ranging from α = 0.91 to 0.94 for informational exclusion and from α = 0.90 to 0.92 for relational exclusion. However, as this is, to our knowledge, the first investigation that used this scale in German, we additionally conducted a confirmatory factor analysis (CFA). Model fit was assessed with five test statistics: (a) the chi-square test statistic (to test if the proposed model provides a plausible structure that can be found in the data), (b) the comparative fit index (*CFI*; an acceptable fit is inferred if the *CFI* is 0.95 or higher), (c) the Tucker–Lewis index (*TLI*; an acceptable fit is inferred if the *TLI* is 0.95 or higher), (d) the root mean square error of approximation (*RMSEA*; an acceptable fit is inferred if the *RMSEA* smaller than 0.08), and (e) the standardized root mean square residual (*SRMR*; an acceptable fit is inferred when the *SRMR* is smaller than 0.08). DWLS was used as estimator as it is the preferred method for assessing ordinal data [64,65]. Instructions provided by Kline [66] were followed to assess these critical values for model fit. The CFA revealed an acceptable fit for t1 (X²(45) = 1363.28, *p* < 0.001, *CFI* = 1.00, *TLI* = 1.01, *RMSEA* < 0.01, and *SRMR* = 0.06) and t2 (X²(45) = 1560.54, *p* < 0.001, *CFI* = 1.00, *TLI* = 1.01, *RMSEA* < 0.01, and *SRMR* = 0.06). The results supported the two-factor structure for the wFoMO scale.

**Workplace telepressure**. Workplace telepressure was measured using six items that assessed the felt pressure of needing to immediately respond to messages in the work context [59]. The scale included the dimensions of *preoccupation* (e.g., it’s hard for me to focus on other things when I receive a message from someone) and *urge* (e.g., I feel a strong need to respond to others immediately) with satisfactory reliability coefficients in this study (α = 0.90–0.92). Moreover, two additional CFAs were conducted to strengthen the evidence that workplace telepressure can be distinguished from wFoMO. The results for t1 (X²(120) = 2527.26, *p* < 0.001, *CFI* = 1.00, *TLI* = 1.03, *RMSEA* < 0.01, and *SRMR* = 0.06) and t2 (X²(120) = 3609.65, *p* < 0.001, *CFI* = 1.00, *TLI* = 1.02, *RMSEA* < 0.01, and *SRMR* = 0.06) revealed an acceptable fit and supported the empirical evidence of wFoMO and workplace telepressure as being two separate constructs [10,32].

**ICT use**. The usage of work-related ICTs was measured with four items. The participants were asked how often they used specific media for work purposes in the first place, namely (1) work-related social media sites such as LinkedIn, (2) e-mails, (3) communication tools like Microsoft Teams, and (4) messenger apps such as WhatsApp.

**Recovery experiences**. General levels of recovery experiences were measured with the Recovery Experience Questionnaire that was originally developed in German by Sonnentag and Fritz [57]. Three dimensions of this questionnaire were used in this study, namely *psychological detachment* (e.g., I forget work), *relaxation* (e.g., I do things that make me relax), and *control* (e.g., I determine my own daily routine). Four items from each dimension were rated, indicating good reliability coefficients for all three subdimensions (α = 0.81–0.93).

**Informational benefits**. Perceived informational benefits were measured using a scale that was developed by Wickramasinghe and Weliwitigoda [58]. Participants rated their level of agreement regarding three subscales: *knowledge sharing* (e.g., I share useful new knowledge that I acquired among my network members), *job openings* and *job security* (e.g., I receive information about job opportunities from my network members), and *career progression* (e.g., the relationships that I maintain are helpful in making career moves). Each subscale showed satisfactory reliability coefficients (α = 0.87–0.89).

**Burnout**. For measuring feelings of burnout, the dimension *work-related burnout* of the Copenhagen Burnout Inventory (CBI) was used [67]. Seven items measured participants’ exhaustion regarding their work (e.g., do you feel worn out at the end of the working day?). Past research achieved satisfactory results regarding reliability, which were supported in this study (α = 0.94–0.95).

**Perceived stress**. For the assessment of perceived stress, the subscale *stress* of the Depression, Anxiety, and Stress Scale (DASS) was utilized [68]. Past research revealed that the DASS is also suitable for use in non-clinical samples [69]. Seven items (e.g., I found it hard to wind down) were answered on a four-point Likert scale ranging from 1 = “Did not apply to me at all” to 4 = “Applied to me very much or most of the time”. Past studies, as well as the current investigation, reported good reliability (α = 0.89–0.90).

### 2.4. Statistical Analyses

The design of this study allowed for us to analyze data from two measuring points that were one week apart. For testing our hypotheses, we employed regression analyses as well as mediational analyses utilizing the software RStudio version 2022.12.0.353 and the package “lavaan”. Before conducting any analysis, we conducted checks regarding normal distribution, Cronbach’s alpha, variance inflation factor, and discriminant validity using the Heterotrait–Monotrait (HTMT) ratio. In the linear regression analyses, data from t1 were used for the independent variables, and data from t2 were used for the dependent variables. The same pattern was maintained for the mediational analyses; however, the data for the mediator variable (ICT use) were also derived from t1. In the mediational analyses, we also controlled for each dependent variable (t2 measure) with t1 data.

## 3. Results

### 3.1. Descriptive Statistics and Correlations

Table 1 and Table 2 summarize descriptive statistics, reliability coefficients, and correlations between all scales for t1 and t2. Further, skew and kurtosis were in an acceptable range [70], as were the maximal values of the HTMT ratio (see Appendix A, Appendix B and Appendix C) [71,72]. For wFoMO, the means for informational exclusion were higher than for relational exclusion at both measuring points. The highest mean was found for burnout with *M* = 3.74–3.76 (*SD* = 0.93–0.96), and the lowest mean was revealed for the dimension relational exclusion of the wFoMO scale with *M* = 1.75–2.05 (*SD* = 0.87–0.93). Overall, no surprising results occurred regarding the descriptive statistics.

According to Cohen [73], effect sizes from *r* = |0.10| are denoted small, from *r* = |0.30| moderate, and from *r* = |0.50| strong. Accordingly, both dimensions of wFoMO as well as telepressure revealed medium to strong positive associations with ICT use. ICT use had the strongest positive association with the outcome variable informational benefits (*r* = 0.56–0.59) and the strongest negative association with psychological detachment (*r* = −0.51–0.53). It was interesting to note that work-related ICT use during leisure time showed significant correlations with almost all of the outcome variables (recovery experiences, perceived stress, burnout, and informational benefits).

### 3.2. Test of Hypotheses

The direct effects of the mediational analyses were used to test the direct associations that were hypothesized in H1a–H2d. All models are summarized in Table 3. Both dimensions of wFoMO as well as telepressure were significantly positively associated with ICT use, supporting *H1a* and *H1b*. Note that we tested for multicollinearity due to the strong associations between these variables (*r* = 0.29–0.59). The variance inflation factor was, however, unproblematic. Further analyses were performed to explore whether ICT use predicted the outcome variables. A significant positive effect was revealed regarding perceived stress (not statistically significant in the model with informational wFoMO as the independent variable) but not regarding burnout, which supports *H2a* but not *H2b*. Regarding recovery experiences, we found evidence for significant negative effects between ICT use and psychological detachment as well as relaxation (only in the model with relational wFoMO as the independent variable) but a non-significant effect between ICT use and the control. *H2c* was deemed to be partially supported. However, a positive significant effect was revealed for ICT use and informational benefits, providing support for *H2d*.

We further conducted mediational analyses to test *H3a–H4d* (note that we thought about the possibility to test the hypotheses by conducting an overall SEM; however, this was not possible due to power issues). To do so, we used either wFoMO (informational and relational exclusion) or telepressure as independent variables, ICT use as the mediator variable, and either perceived stress, burnout, recovery experiences, or informational benefits as dependent variables. Indirect effects were rated as significant if the confidence interval did not include zero [74]. Significant indirect effects were revealed for both wFoMO subdimensions regarding perceived stress (informational exclusion: 0.06, *SE* = 0.05, and 95% *CI* [0.000, 0.114]; relational exclusion: 0.12, *SE* = 0.05, and 95% *CI* [0.032, 0.217]). *H3a* was supported. The indirect effect with burnout as the dependent variable was, however, only significant for relational exclusion wFoMO (0.07, *SE* = 0.03, and 95% *CI* [0.001, 0.120]) but not informational exclusion wFoMO (0.04, *SE* = 0.03, and 95% *CI* [−0.025, 0.084]). Therefore, *H3b* was only partially supported. We conducted separate mediation analyses to distinguish between psychological detachment, relaxation, and control as the dependent variables. Regarding psychological detachment, the indirect effects were significant for both wFoMO dimensions (informational exclusion: −0.19, *SE* = 0.08, and 95% *CI* [−0.390, −0.059]; relational exclusion: −0.34, *SE* = 0.15, and 95% *CI* [−0.811, −0.173]). Regarding relaxation, the indirect effect was only significant for relational exclusion (−0.09, *SE* = 0.05, and 95% *CI* [−0.248, −0.018]) but not informational exclusion (−0.06, *SE* = 0.05, and 95% *CI* [−0.174, 0.024]). In contrast, regarding the control, both indirect effects remained non-significant (informational exclusion: −0.00; *SE* = 0.05, and 95% *CI* [−0.101, 0.080]; relational exclusion: −0.04, *SE* = 0.05, and 95% *CI* [−0.144, 0.042]). Therefore, *H3c* was only partially supported. Lastly, we tested whether there was an indirect effect with informational benefits as the outcome variable. We found evidence for *H4d*, since the indirect effect became significant for both wFoMO dimensions (informational exclusion: 0.27, *SE* = 0.09, and 95% *CI* [0.123, 0.469]; relational exclusion: 0.25, *SE* = 0.10, and 95% *CI* [0.126, 0.560]).

The same mediational analyses were conducted again, this time using workplace telepressure as the independent variable. The indirect effect from workplace telepressure via ICT use on perceived stress was significant (0.08, *SE* = 0.03, and 95% *CI* [0.793, 0.983]), which provided support for *H4a*. The indirect effect regarding burnout was not significant (0.03, *SE* = 0.06, and 95% *CI* [−0.025, 0.077]). *H4b* was not supported. Regarding recovery experiences, different results were found depending on the specific subdimension. The indirect effect on psychological detachment was significant (−0.20, *SE* = 0.08, and 95% *CI* [−0.409, −0.085]). However, it was not significant for relaxation (−0.06, *SE* = 0.05, and 95% *CI* [−0.171, 0.008]) and the control (−0.03, *SE* = 0.04, and 95% *CI* [−0.117, 0.041]), indicating only partial evidence for *H4c.* Similar to the results for both wFoMO dimensions, there was a significant indirect effect on informational benefits (0.31, *SE* = 0.07, and 95% *CI* [0.133, 0.645]), providing evidence for *H4d*. See Table 3 and Figure 2, Figure 3 and Figure 4 for an overview (see Appendix D for the specific determination coefficients).

## 4. Discussion

The current study demonstrates that the experience of wFoMO and workplace telepressure is associated with employee well-being variables mediated via ICT use during leisure time after a typical work week. These findings will be discussed in more detail in the following sections.

In accordance with the past literature [8,10,32], the hypotheses that wFoMO and workplace telepressure have a positive direct effect on after-workhour ICT use have been supported. In this context, the question arises as to whether wFoMO and workplace telepressure are distinguishable constructs. From a theoretical perspective, both constructs can be understood as a state of mind. For workplace telepressure, this directly relates to ICTs, as it is defined as the constant need to check work-related messages [8]. The experience of wFoMO, however, is considered as the fear of missing out on important work-related information or contacts [10]. However, we have shown that ICTs also play a crucial role when dealing with feelings of wFoMO, which is in line with studies focusing on general FoMO [13]. Moreover, the study by Budnick et al. [10], which is the only study we know that used the same assessment of wFoMO as we carried out, showed that wFoMO and workplace telepressure are both associated with variables like message checking behavior or social media engagement. However, differential validity has been revealed for personality variables. Significant negative associations were explored between wFoMO, agreeableness, and consciousness, and a significant positive association between wFoMO and neuroticism. No such significant association was found regarding workplace telepressure [10]. Although future studies should try to replicate these findings, the current study also strengthens that the constructs workplace telepressure and wFoMO are related but must be distinguished. In the context of these findings, correlational analyses in this study showed that medium to strong associations can be found between both constructs (*r* = 0.29–0.53); therefore, additional CFAs were conducted for both measurement times, yielding an acceptable fit and strengthening previous empirical evidence of both constructs being distinguishable [10,31].

We further investigated direct effects between ICT use and employee well-being and indirect effects between wFoMO as well as workplace telepressure and employee well-being mediated via ICT use. We found a significant direct positive association between ICT use and perceived stress, as well as significant indirect effects. In particular, wFoMO and workplace telepressure showed a positive effect on perceived stress via ICT use. Regarding burnout, the results were more ambivalent. We did not find a direct significant effect between ICT use and burnout, and there only was one significant indirect effect for the subdimension of relational exclusion wFoMO on burnout via ICT use. However, these mostly non-significant findings do not mean that there are, in general, no significant associations [75]. For example, Budnick et al. [10] found a significant association between message checking behavior as well as media engagement and burnout in a cross-sectional study, but used a different instrument to measure burnout than we did in this study. In addition, the one-week period could have been too short to measure the impact of ICT use on burnout. It may simply take longer for effects on burnout to be measurable. Therefore, future studies should still consider burnout as an important outcome while using adapted methods and/or longer time intervals.

Regarding the relationship between recovery experiences and ICT use, the results differed between the recovery experiences. While significant negative associations for ICT use were revealed for psychological detachment and relaxation, the association between control and ICT use remained non-significant. The findings for psychological detachment and relaxation are in line with previous studies and strengthens the assumption that work-related ICT use during leisure time is considered as a job demand [44]. Moreover, the anticipated negative indirect effects regarding psychological detachment were revealed for all independent variables, i.e., informational and relational exclusion wFoMO and workplace telepressure. For relaxation, a significant negative indirect effect was only revealed for the wFoMO subdimension of relational exclusion. No significant indirect effects were revealed for the control. It was surprising that neither the direct effect nor the indirect effects for the recovery experience control were observed. A possible explanation may be that even after work hours, ICT use does not necessarily go hand in hand with a perceived loss of control but rather harms detachment and relaxation processes.

Furthermore, the anticipated positive association between ICT use and informational benefits became significant. In addition, the assumed positive indirect effects for informational benefits also reached statistical significance across all independent variables. This finding strengthens the previous literature that discussed the ambivalence of work-related ICT use being both a job demand, and a job resource [48]. However, this study is one of the first to highlight that this ambivalence must also be considered when it comes to work-related ICT use during leisure time.

Due to empirical evidence [9] and the JD-R theory, we only assumed indirect effects and no full mediation. For most of the analyses, this assumption was confirmed in the results of this study, as no significant direct effects between the independent and the dependent variables were observed, whereas the indirect effect reached significance. However, for some analyses, this was not the case, and the direct effect became significant, namely for (1) relational exclusion wFoMO → burnout, (2) relational exclusion wFoMO → psychological detachment, (3) informational exclusion wFoMO → informational benefits, and (4) workplace telepressure → informational benefits. It is up to future studies to show whether these effects are stable. Notably, the negative direct effects regarding informational benefits are of particular interest. The initially negative effect became positive when ICT use was included as a mediator. On the one hand, it strongly reinforces the positive influence of ICT use. On the other hand, it shows that wFoMO and workplace telepressure are personal demands that may indeed have negative effects that need to be coped with.

### 4.1. Theoretical and Practical Implications

From a theoretical perspective, it may be fruitful to take a closer look at the construct of wFoMO and how it arises. As stated before, the constructs of workplace telepressure and wFoMO differ. While the definition of Barber and Santuzzi [8] indicates that ICTs are ubiquitous to experience workplace telepressure, it is unclear as to whether the existence of ICTs is also needed to experience wFoMO. This consideration of ICTs not always being essential for consequences of wFoMO to emerge is strengthened by the significant direct effects between informational and relational wFoMO and the outcome variables revealed in this study. In line with our reasoning for understanding wFoMO as a personal demand, it could be that for employees who experience very high levels of wFoMO, direct associations between wFoMO and work-related well-being occur. This needs further research to answer the question of how ICT use during working hours or during leisure time is crucial to experiencing wFoMO. In the context of this, it is also questionable as to whether the definition of Budnick et al. [10] holds, as they stated that one must be “away or disconnected from work” to experience wFoMO, which is one reason why we focused on the after-workhour effects of ICT use on employees in the first place. This definition indicates that employees who work from home or elsewhere are more likely to experience wFoMO. However, as we see wFoMO as a personal demand, it could also be that this arises not solely from the working environment but also differs between individuals on a more general level. It is an open question as to whether employees with high trait levels of wFoMO experience, in general, more anxiety of missing out on something, even though they may be present at the workplace. At the same time, this would suggest that for employees with a lower wFoMO level, situational factors are more likely to determine whether the experience of wFoMO rises. Such a trait vs. state discussion has already been held for general FoMO [76] and is not finished yet. An experience sampling study with (1) measurements during and after working hours and (2) on days at which employees work at the workplace vs. remotely could help to identify whether wFoMO can be seen rather as a trait, a state, or both, and whether being away from work is crucial to experiencing wFoMO.

In view of the fact that the working environment and, thus, the demands on the workplace are changing rapidly as a result of digitization and the possibility of working independently of the workplace, it is important to take the resulting new personal demands into account. By arguing with the JD-R theory, wFoMO and workplace telepressure are clearly perceived as personal demands. ICT use during leisure time may serve as a strategy to cope with these demands. However, the prior literature is ambiguous when it comes to the categorization of ICT use as a job demand vs. a job resource [49]. Empirical evidence by Budnick et al. [10] and our study highlight the reasoning that wFoMO is another relevant construct that should be considered when exploring personal demands at the workplace. However, this is only relevant due to the impact the experience of wFoMO has on further work-related well-being variables mediated via ICT use. This study has shown that this impact is rather negative, especially for the variables of perceived stress and psychological detachment. Nevertheless, we were able to show that despite these negative effects, after-workhour ICT use can also be seen as a job resource due to enhanced perceived informational benefits. In this regard, a study by Ninaus et al. [48] must be mentioned, as they also explored whether ICT use is seen as a job demand vs. a job resource. They found that although employees perceived ICTs as a job resource rather than a job demand, strong negative effects occurred, like work–family imbalance, increased levels of burnout, and reduced job satisfaction [48]. They further concluded that, therefore, “the premise of the JD-R model that job resources lead to positive outcomes or reduce unfavorable effects of working life [39,40] may not hold in the context of ICTs” [48]. Our study appears to be a fruitful extension to this conclusion, as we did, indeed, find a variable, namely informational benefits, which was positively affected by ICT use during leisure time.

Nevertheless, the negative consequences revealed in this study should not be ignored. We have shown that especially psychological detachment and perceived stress, and for some subdimensions of wFoMO, relaxation and burnout, are harmed by after-workhour ICT use due to wFoMO and workplace telepressure. Previous studies have already revealed that a lack of recovery experiences and high levels of perceived stress can lead to further negative outcomes [77]. Often, these do not only harm employee well-being but also organizational outcomes due to fluctuations or reduced job performance [78,79]. It is up to future studies to explore whether organizational outcomes can also be directly linked to wFoMO.

### 4.2. Strengths, Limitations, and Future Research

The design with two measurement points allowed for us to test how wFoMO, workplace telepressure, and ICT use during leisure time affect work-related well-being outcomes after a regular working week. We chose the time interval of one work week due to the following reasons: (1) At t1, the participants answered all the variables on a general level. However, at t2, we asked the participants to specifically answer the same items in relation to their last working week. This allowed for us to control all of the outcome variables assessed at t2 for their baseline level at t1. (2) There were (almost) no theoretical considerations and only little evidence, especially for wFoMO, on what intervals are appropriate for measuring effects. From general media research, we have concluded that effects may be best measured by studying relatively short intervals. (3) In the context of (2), we decided that the interval of one work week was a good starting point for further research into the relatively new construct of wFoMO. It is up to future studies to investigate whether these effects are still significant with longer intervals between measurements.

Furthermore, we measured wFoMO, workplace telepressure, and ICT use at the same time, as past studies have shown that these constructs often occur together [13]. Moreover, we assessed these constructs in light of the JD-R theory and considered them as relatively stable personal demands or a job demand/resource. However, it could also be that the levels of wFoMO and workplace telepressure, and, therefore, subsequent ICT use, vary depending on special events or situational factors at work. Future research could answer this question by employing a daily diary approach.

Another strength of the design with two paired measurement points is that we assessed all the outcome variables at t1 and t2. Therefore, we were able to control the outcome variables at t2 for t1 levels, resulting in more convincing analyses. Although this design still does not allow us to draw causal conclusions (it would need experimental conditions to perform this), controlling for t1 levels provides evidence that there are robust associations.

In this context, a limitation is that this design does not answer the question of whether wFoMO and workplace telepressure or ICT use are predicting one another. There are multiple studies that either assumed that wFoMO and workplace telepressure predict ICT use [9] or vice versa [80]. However, this question could be extremely difficult to answer, as there may also be reciprocal effects. As we perceived wFoMO and workplace telepressure as personal demands and ICT use as a job demand/job resource in light of the JD-R theory, it was reasonable to specify ICT use as the mediator. Nevertheless, we are aware that this may not reflect the full picture.

The next limitation relates to the sample size of 130 employees. However, post-hoc analyses revealed that a sample size of 126 people would be sufficient to detect even small effects when conducting multiple linear regression with three predictors (*f*² = 0.02, α = 0.05, *n* = 130). In addition, we employed bootstrapping to analyze the indirect effects, which is an appropriate procedure for relatively small samples. However, to strengthen the significance of the results, future studies should strive for larger sample sizes. Therefore, future studies should also try to achieve more balanced samples in terms of gender in order to increase generalizability. In the context of the sample size, there is another limitation to consider. When evaluating structural equation models, it is useful to assess and interpret the determination coefficient, as well as the effect size [81]. However, we could not report on the latter, as we tested each variable independently in the first place. It would be beneficial for future studies to consider this aspect, as past studies have yielded fruitful results when doing so [81].

Another point that needs to be discussed is the usage of burnout as an outcome variable. While we did find stable effects for all hypotheses regarding stress, many effects remained non-significant for burnout. As indicated, this could be due to measurement issues and should be further explored in upcoming studies.

Another measurement issue is that we solely assessed work-related ICT use as well as wFoMO and workplace telepressure. However, in daily usage behaviors, private and work-related media engagement may be blurred. For example, some people may use the same devices (e.g., smartphones and tablets) to check on private and work-related e-mails or social media sites. Therefore, spillover effects from work-related to private media behavior and vice versa cannot be ruled out and should be considered in future investigations. By doing so, the effects of professional ICT use, or wFoMO, and workplace telepressure could be controlled for the private counterpart.

## 5. Conclusions

We extended prior work that discussed the question of whether work-related ICT use is perceived as a job demand vs. a job resource in light of the JD-R theory by exploring ICT usage behaviors outside regular working hours. As expected, we revealed multiple negative effects on perceived stress and psychological detachment. However, even ICT use during leisure time seems to have positive effects when considering perceived informational benefits. Furthermore, we argued that wFoMO and workplace telepressure can be integrated as personal demands in the JD-R theory, which was supported by significant indirect effects between wFoMO as well as workplace telepressure and employee well-being outcomes via work-related ICT use during leisure time in this study. In a time of digitalization and flexible working models, it becomes more and more important to reveal the consequences of new personal demands, such as wFoMO, that arise from this development.

## Figures and Tables

**Figure 1 behavsci-14-00008-f001:**
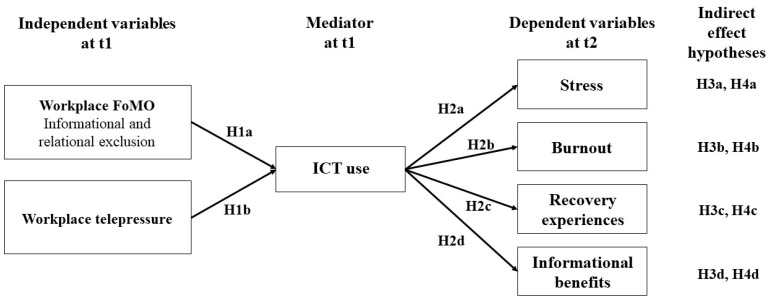
Overview of all the hypotheses.

**Figure 2 behavsci-14-00008-f002:**
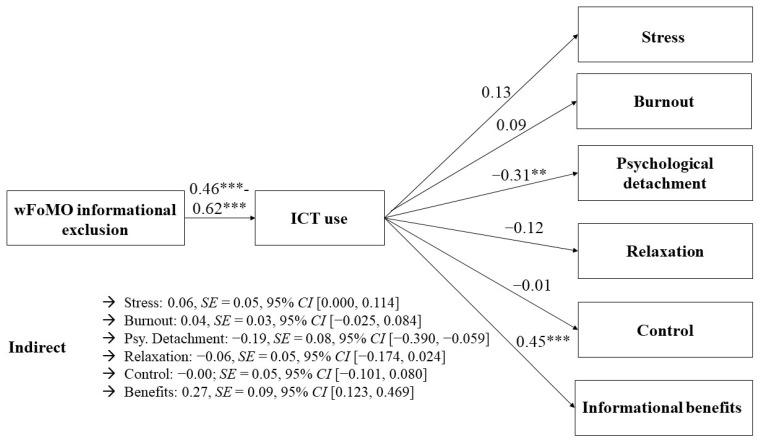
Associations between informational exclusion wFoMO and employee well-being mediated via ICT use. ** *p* < 0.01. *** *p* < 0.001.

**Figure 3 behavsci-14-00008-f003:**
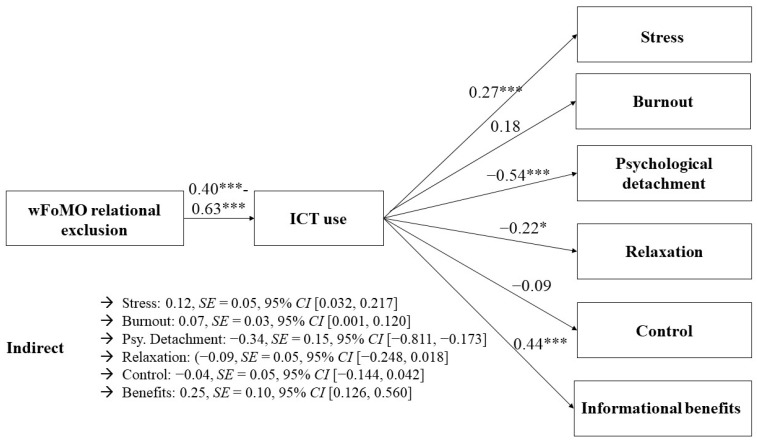
Associations between relational exclusion wFoMO and employee well-being mediated via ICT use. * *p* < 0.05. *** *p* < 0.001.

**Figure 4 behavsci-14-00008-f004:**
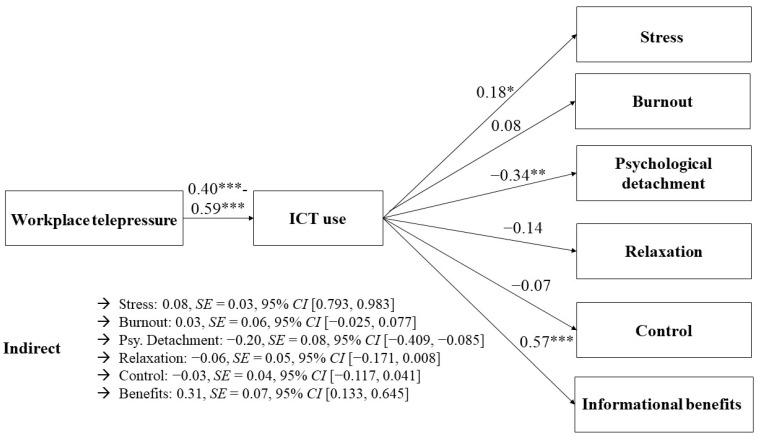
Associations between workplace telepressure and employee well-being mediated via ICT use. * *p* < 0.05. ** *p* < 0.01. *** *p* < 0.001.

**Table 1 behavsci-14-00008-t001:** Means, standard deviations, and reliability coefficients for all scales at t1 and t2.

	t1			t2		
Variable	*M*	*SD*	α	*M*	*SD*	α
wFoMO						
Informational	2.58	1.02	0.91	2.40	1.08	0.94
Relational	2.05	0.93	0.90	1.75	0.87	0.92
Telepressure	3.21	0.98	0.90	2.99	1.03	0.92
ICT use	2.61	1.07	0.86	2.46	0.99	0.86
Recovery experiences						
Detachment	2.11	0.60	0.83	2.08	0.59	0.81
Relaxation	2.14	0.71	0.89	2.08	0.75	0.91
Control	3.16	0.93	0.90	3.38	0.94	0.93
Stress	3.34	0.94	0.89	3.47	0.94	0.90
Burnout	3.74	0.93	0.94	3.76	0.96	0.95
Benefits	3.09	0.84	0.87	2.83	0.93	0.89

Note. wFoMO = workplace fear of missing out; and ICT = information and communication technology.

**Table 2 behavsci-14-00008-t002:** Intercorrelations for t1 and t2.

Variable	1	2	3	4	5	6	7	8	9	10
wFoMO										
1. Informational	-	0.48 ***	0.39 ***	0.46 ***	−0.42 ***	−0.25 **	−0.23 **	0.52 ***	0.23 **	0.33 ***
2. Relational	0.59 ***	-	0.29 ***	0.38 ***	−0.21 *	−0.09	−0.07	0.28 **	0.20 *	0.29 ***
3. Telepressure	0.53 ***	0.52 ***	-	0.39 ***	−0.42 ***	−0.32 ***	−0.16	0.45 ***	0.24 **	0.24 **
4. ICT use	0.59 ***	0.51 ***	0.47 ***	-	−0.53 ***	−0.25 **	−0.17 *	0.29 ***	0.13	0.59 ***
Recovery experiences										
5. Detachment	−0.48 ***	−0.24 **	-0.35***	−0.51 ***	-	0.50 ***	0.32 ***	−0.49 ***	−0.41 ***	−0.27 **
6. Relaxation	−0.26 **	−0.10	−0.17	−0.22 *	0.61 ***	-	0.64 ***	−0.46 ***	−0.43 ***	−0.15
7. Control	−0.19 *	−0.12	0.14	−0.13	0.36 ***	0.62 ***	-	−0.37 ***	−0.33 ***	−0.01
8. Stress	0.51 ***	0.24 **	0.41 ***	0.34 ***	−0.62 ***	−0.54 ***	−0.36 ***	-	0.63 ***	0.06
9. Burnout	0.28 **	0.04	0.26 **	0.14	−0.47 ***	−0.45 ***	−0.37 ***	0.63 ***	-	−0.10
10. Benefits	0.44 ***	0.44 ***	0.29 ***	0.56 ***	−0.29 ***	−0.18 *	−0.08	0.19 *	−0.03	-

Note. wFoMO = workplace fear of missing out; and ICT = information and communication technology. The results for t1 are shown above the diagonal. The results for t2 are shown under the diagonal. * *p* < 0.05. ** *p* < 0.01. *** *p* < 0.001.

**Table 3 behavsci-14-00008-t003:** Results of all mediational analyses for *H1a*–*H4d*.

				95% *CI*		
	β	*b*	*SE*	*LL*	*UL*	*p*-Value
	**Stress**					
wFoMO inf → ICT use	0.476	0.505	0.081	0.332	0.655	<0.001
ICT use	0.131	0.091	0.051	−0.002	0.197	0.077
Direct effect	0.010	0.007	0.062	−0.126	0.115	0.906
Indirect effect	0.062	0.046	0.029	0.000	0.114	0.110
Total effect	0.868	0.886	0.050	0.787	0.982	<0.001
wFoMO rel → ICT use	0.447	0.526	0.112	0.292	0.731	<0.001
ICT use	0.265	0.184	0.064	0.066	0.321	0.004
Direct effect	−0.118	−0.096	0.087	−0.308	0.043	0.268
Indirect effect	0.118	0.097	0.045	0.032	0.217	0.031
Total effect	0.835	0.874	0.058	0.763	0.988	<0.001
Telepressure → ICT use	0.432	0.479	0.103	0.266	0.674	<0.001
ICT use	0.181	0.126	0.054	0.029	0.240	0.019
Direct effect	−0.021	−0.016	0.070	−0.173	0.101	0.816
Indirect effect	0.078	0.060	0.033	0.013	0.145	0.070
Total effect	0.864	0.889	0.049	0.793	0.983	<0.001
	**Burnout**					
wFoMO inf → ICT use	0.464	0.490	0.078	0.323	0.631	<0.001
ICT use	0.085	0.047	0.053	−0.057	0.152	0.378
Direct effect	−0.003	−0.002	0.055	−0.110	0.102	0.975
Indirect effect	0.039	0.023	0.027	−0.025	0.084	0.402
Total effect	0.789	0.766	0.054	0.653	0.864	<0.001
wFoMO rel → ICT use	0.398	0.462	0.104	0.246	0.651	<0.001
ICT use	0.175	0.096	0.051	−0.004	0.200	0.061
Direct effect	−0.190	−0.121	0.055	−0.245	−0.027	0.028
Indirect effect	0.070	0.045	0.029	0.001	0.120	0.119
Total effect	0.667	0.702	0.054	0.588	0.801	<0.001
Telepressure → ICT use	0.397	0.437	0.099	0.228	0.622	<0.001
ICT use	0.076	0.042	0.054	−0.062	0.148	0.437
Direct effect	0.044	0.027	0.052	−0.078	0.126	0.608
Indirect effect	0.030	0.018	0.025	−0.025	0.077	0.474
Total effect	0.817	0.780	0.055	0.665	0.882	<0.001
	**Psychological detachment**					
wFoMO inf → ICT use	0.624	0.692	0.087	0.513	0.859	<0.001
ICT use	−0.305	−0.267	0.102	−0.482	−0.082	0.009
Direct effect	0.165	0.160	0.143	−0.073	0.497	0.264
Indirect effect	−0.190	−0.185	0.082	−0.390	−0.059	0.025
Total effect	0.694	0.703	0.151	0.440	1.035	<0.001
wFoMO rel → ICT use	0.625	0.786	0.148	0.473	1.060	<0.001
ICT use	−0.537	−0.471	0.121	−0.774	−0.284	<0.001
Direct effect	0.521	0.575	0.253	0.217	1.256	0.023
Indirect effect	−0.336	−0.370	0.151	−0.811	−0.173	0.014
Total effect	1.011	1.039	0.251	0.700	1.561	<0.001
Telepressure → ICT use	0.588	0.688	0.119	0.450	0.917	<0.001
ICT use	−0.336	−0.295	0.089	−0.481	−0.124	0.001
Direct effect	0.212	0.217	0.143	−0.013	0.556	0.129
Indirect effect	−0.198	−0.203	0.079	−0.409	−0.085	0.010
Total effect	0.757	0.765	0.165	0.465	1.117	<0.001
	**Relaxation**					
wFoMO inf → ICT use	0.496	0.529	0.080	0.365	0.680	<0.001
ICT use	−0.123	−0.107	0.088	−0.293	0.052	0.224
Direct effect	0.017	0.015	0.089	−0.139	0.210	0.862
Indirect effect	−0.061	−0.057	0.050	−0.174	0.024	0.253
Total effect	0.628	0.629	0.113	0.414	0.851	<0.001
wFoMO rel → ICT use	0.428	0.502	0.118	0.262	0.720	<0.001
ICT use	−0.220	−0.192	0.090	−0.378	−0.024	0.034
Direct effect	0.129	0.132	0.103	−0.043	0.375	0.203
Indirect effect	−0.094	−0.096	0.054	−0.248	−0.018	0.077
Total effect	0.726	0.725	0.112	0.509	0.950	<0.001
Telepressure → ICT use	0.430	0.477	0.098	0.273	0.659	<0.001
ICT use	−0.144	−0.126	0.083	−0.298	0.023	0.130
Direct effect	0.092	0.089	0.102	−0.091	0.309	0.384
Indirect effect	−0.062	−0.060	0.045	−0.171	0.008	0.181
Total effect	0.727	0.724	0.137	0.462	1.009	<0.001
	**Control**					
wFoMO inf → ICT use	0.474	0.503	0.078	0.340	0.648	<0.001
ICT use	−0.008	−0.007	0.089	−0.184	0.162	0.934
Direct effect	0.007	0.006	0.086	−0.145	0.193	0.942
Indirect effect	−0.004	−0.004	0.046	−0.101	0.080	0.936
Total effect	0.740	0.760	0.098	0.567	0.950	<0.001
wFoMO rel → ICT use	0.404	0.470	0.109	0.245	0.668	<0.001
ICT use	−0.091	−0.081	0.093	−0.271	0.093	0.385
Direct effect	0.067	0.069	0.100	−0.110	0.284	0.488
Indirect effect	−0.037	−0.038	0.046	−0.144	0.042	0.408
Total effect	0.770	0.791	0.094	0.604	0.973	<0.001
Telepressure → ICT use	0.410	0.453	0.097	0.247	0.630	<0.001
ICT use	−0.067	−0.059	0.080	−0.223	0.093	0.456
Direct effect	0.085	0.083	0.079	−0.056	0.0259	0.291
Indirect effect	−0.027	−0.027	0.038	−0.117	0.041	0.483
Total effect	0.806	0.825	0.093	0.640	1.008	<0.001
	**Informational benefits**					
wFoMO inf → ICT use	0.608	0.670	0.094	0.474	0.846	<0.001
ICT use	0.449	0.389	0.103	0.196	0.600	<0.001
Direct effect	−0.250	−0.239	0.134	−0.554	−0.020	0.075
Indirect effect	0.273	0.260	0.087	0.123	0.469	0.003
Total effect	0.776	0.853	0.060	0.729	0.965	<0.001
wFoMO rel → ICT use	0.578	0.711	0.147	0.413	0.984	<0.001
ICT use	0.438	0.380	0.101	0.194	0.591	<0.001
Direct effect	−0.213	−0.228	0.167	−0.672	0.023	0.173
Indirect effect	0.253	0.270	0.104	0.126	0.560	0.010
Total effect	0.785	0.865	0.063	0.736	0.982	<0.001
Telepressure → ICT use	0.552	0.636	0.140	0.353	0.905	<0.001
ICT use	0.570	0.494	0.117	0.283	0.745	<0.001
Direct effect	−0.422	−0.422	0.194	−0.905	−0.127	0.030
Indirect effect	0.314	0.314	0.127	0.133	0.645	0.014
Total effect	0.702	0.787	0.071	0.645	0.922	<0.001

Note. wFoMO = workplace fear of missing out; ICT = information and communication technology; β = standardized regression coefficient; *b* = regression coefficient; *CI* = confidence interval; *LL* = lower limit; and *UL* = upper limit.

## Data Availability

The dataset generated and analyzed during the current study can be reviewed at: https://doi.org/10.17605/OSF.IO/YXTVZ (accessed on 9 September 2023).

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
