# Peer review of "I Do Not Want to Miss a Thing! Consequences of Employees’ Workplace Fear of Missing Out for ICT Use, Well-Being, and Recovery Experiences"

_behavsci, 2023, doi:10.3390/bs14010008_

Round 1

Reviewer 1 Report

Comments and Suggestions for Authors

Thank you very much for opportunity to read this interesting paper.I think it is a well-structured and very interesting work. I hope that these comments, offered with a constructive spirit, prove useful for further developing your work.

1) I would perform attrition analysis and report the results.

2) ethical considerations. there are no references to data protection law

Reviewer 2 Report

Comments and Suggestions for Authors

The work presents an interesting theme and is generally well presented and documented, both at the theoretical level and at the methodological and empirical level. I believe that the introduction can be improved, particularly in the presentation of the concepts under analysis and in the relevance of the study. I consider that this statement is unnecessary in the introduction (p. 48–50).

"We conducted a study with N = 130 employees that participated in two measuring points at the interval of one typical working week. This design enables us to trace how wFoMO and workplace telepressure sensations, mediated by ICT use, influence key work-related outcomes after a workweek."

I consider that the hypotheses set out could be highlighted in the text, giving them greater visibility.

With regard to the statement on p. 322–323, I consider that "To address possible participants, flyers were distributed, e-mails were sent, and social media sites like LinkedIn, Instagram, and WhatsApp were utilized." It should be in the procedure section instead of the participant.

I was in doubt if the design of the study can be considered longitudinal; although it was a measurement in two moments obeying a plan of repeated measurements, I believe that the time period is too short for this designation, so the authors will be able to better verify this information.

Reviewer 3 Report

Comments and Suggestions for Authors

Dear Authors,

Thank you for the opportunity to review an interesting article entitled: “I don’t want to miss a thing! Consequences of Employees’ Workplace Fear of Missing Out for ICT Use, Well-Being, and Recovery Experiences.” The aim of this article is to investigate whether Feelings of workplace Fear of Missing Out (wFoMO) and workplace telepressure are associated with employee well-being variables via the use of ICTs during leisure time. This topic is of a great interest to both FoMO professionals and ICTs industry. Overall this article will make a contribution to the literature that can be used and cited by future researchers. However, the following comments, which aim to improve the text, are worth considering:

Comment 1: Introduction. Line 40. The author/s cited Budnick et al. (2020) 18 times, but can’t find it in the References.

Comment 2: Consequences of wFoMO, telepressure, and ICT use. Line 285. I suggest the author/s to list these hypotheses separately along with supporting arguments to make it clear for readers.

Comment 3: Consequences of wFoMO, telepressure, and ICT use. Line 292. “Ther is a positive indirect effect between telepressure…” To be consistent with H4a, workplace telepressure is expected here.

Comment 4: Procedure. Line 315. Please elaborate the reasons of adopting one week as proper time interval between t1 and t2.

Comment 5: Participants. Line 327. “… 130 participants with 50 males and 80 females.” The sample size is small. More samples are expected to increase statistically soundness. The author/s may also consider adopting SmartPLS. Besides, the distribution of sex is skewed. The sex could be a possible characteristic affecting research results.

Comment 6: Figure 2,3,4. Line 506. The author/s should list titles beneath figures.

Comment 7: Reference: There are some problems about the references, such as inconsistent capitalization of titles, inconsistent format or incomplete publication information.

Reviewer 4 Report

Comments and Suggestions for Authors

Dear editor

Dear authors,

Thank you for inviting me to review this interesting manuscript.

Im agreed with author that As more and more employees have access to work-related information and communication technologies (ICT) anywhere and anytime, new challenges arise in terms of well-being and recovery experiences.

I believe that this paper have significant contributions in technology field.

Below author can find my comments to improve the quality of manuscript before publications.

Abstract section: the contributions, implications and recommendations of this study can be added.

Introduction: the authors not provide contribution of their study, novelty research and the explanation of the structure of paper in the last paragraph in introduction section

Literature review section is missing

Figure 1: is not clear where is H1 and H2 in the figure? the complete hypothesis must be shown here.

Methodology section: I am afraid that the sample of this study is too small and have a bias.

Descriptive results should include testing the normality of the data. Skewness and kurtosis checks are needed. And Author may add fornell-lacker test or HTMT for discriminant validity. See this papers for references.

“Applying the UTAUT Model to Understand Factors Affecting Micro-Lecture Usage by Mathematics Teachers in China,” Mathematics, vol. 10, no. 7, pp. 1–20, 2022. https://doi.org/10.3390/math10071008

references in this paper are very lacking. This raises the question whether this paper builds based on previous research? and supported by previous research studies? author can add at least 20-30 more references. a good paper requires a strong reference sources.

Round 2

Reviewer 3 Report

Comments and Suggestions for Authors

I have no further suggestions.

Author Response

Thank you for your help and your final decision!

Reviewer 4 Report

Comments and Suggestions for Authors

dear editor

dear authors,

authors has been revised the manuscript but I cannot see the revise version for this comment:

the results should include testing the normality of the data. Skewness and kurtosis checks are needed. And Author may add fornell-lacker test or HTMT for discriminant validity. See this papers for references.

“Applying the UTAUT Model to Understand Factors Affecting Micro-Lecture Usage by Mathematics Teachers in China,” Mathematics, vol. 10, no. 7, pp. 1–20, 2022. https://doi.org/10.3390/math10071008

in additions, According to paper “Predicting Factors Influencing Preservice Teachers ’ Behavior Intention in the Implementation of STEM Education Using Partial Least Squares Approach,” Sustain., 2022. In structural model evaluation, an analysis of the F2 value alsoneeds to be performed. This construct explains the effect of exogenous on endogenous variables to determine changes in the R2 value when specific exogenous determinants are excluded from the model. besides, Moderating Effect Analysis of Gender and Age also can be analysis. 

please revise the manuscript carefully. after auhtor revise the manuscript I believe I can endorse this manuscript.

well done

Round 3

Reviewer 4 Report

Comments and Suggestions for Authors

dear editor,

dear authors,

i have read the revision version of this paper.

now this paper can be accepted for publications.

well done

Author Response

Thank you for all of your suggestions and your final acceptance!